# A pilot randomized controlled trial to explore the feasibility of a peer-delivered single-session brief intervention for youth with moderate risk substance use

Florence Jaguga[1]*, Matthew Turissini[2], Edith Kamaru Kwobah[1†], Edith Apondi[3], Leslie A. Enane[4,5], Julius Barasa[6], Gilliane Kosgei[6], Yvonne Olando[7], Mary A. Ott[8], Allan Kimaina[6], Matthew C. Aalsma[9]

1 Directorate of Mental Health and Rehabilitative Services, Moi Teaching & Referral Hospital, Eldoret, Kenya, 2 Department of Internal Medicine, Indiana University, Indianapolis, Indiana, United States of America, 3 Department of Child Health and Pediatrics, Moi Teaching & Referral Hospital, Eldoret, Kenya, 4 Department of Pediatrics, The Ryan White Center for Pediatric Infectious Disease and Global Health, Indiana University School of Medicine, Indianapolis, Indiana, United States of America, 5 Indiana University Center for Global Health, Indianapolis, Indiana, United States of America, 6 Academic Model Providing Access to Healthcare, Eldoret, Kenya, 7 National Authority for the Campaign against Alcohol and Drug Abuse, Nairobi, Kenya, 8 Department of Global Health and Health System Design and of Pediatrics, Arnhold Institute for Global Health, Icahn School of Medicine at Mount Sinai, New York, New York, United States of America, 9 Department of Pediatrics, Division of Child Health Services Research, Indiana University, Indianapolis, Indiana, United States of America

† Deceased.
* flokemboi@gmail.com

## Abstract

### Background

Youth in sub-Saharan Africa are at high risk of substance use yet lack access to appropriate interventions. The goal of this project was to evaluate the feasibility of a definitive trial to explore efficacy of a peer-delivered single-session brief intervention (SSBI) for youth with substance use in Kenya.

### Methods

Seventy youth aged 15−24 years with moderate risk substance use were randomized to SSBI or to psychoeducation. Data was collected at baseline and month three. *Primary outcomes:* Feasibility criteria, e.g., study participation rate, proportion of participants willing to be randomized, and study completion rate. Strategies for recruitment in a future trial were collected using focus group discussions with the youth at month three. *Secondary outcomes:* (i) Change in substance use (Alcohol, Smoking & Substance Use Involvement Screening Test for Youth [ASSIST-Y] questionnaire), depression (Patient Health Questionnaire [PHQ-9]), anxiety (Generalized Anxiety Disorder [GAD-7 scale]), and quality of life (World Health Organization-Quality of Life Brief

**Data availability statement:** All relevant data are within the paper and its Supporting Information files.

**Funding:** This project was funded, in part, with support from the Indiana Clinical and Translational Sciences Institute funded, in part by Grant Number UL1TR002529 from the National Institutes of Health, National Center for Advancing Translational Sciences, Clinical and Translational Sciences Award. The funders had no role in study design, data collection and analysis, decision to publish, or preparation of the manuscript. There was no additional external funding received for this study.

**Competing interests:** The authors have declared that no competing interests exist.

**Abbreviations:** AMPATH, Academic Model Providing Access to Health Care; ANOVA, Analysis of Variance; ASSIST-Y, Alcohol Smoking and Substance Involvement Screening Test – Youth; BIs, Brief Intervention; CFIR, Consolidated Framework for Implementation Research; CI, Confidence Interval; DALY, Disability Adjusted Life Year; FGD, Focus Group Discussion; FRAMES, Feedback Responsibility Advice Menu of options Empathy Self-efficacy; GAD, Generalized Anxiety Disorder; GAD-7, Generalized Anxiety Disorder – 7; GBD, Global Burden of Disease; HIV, Human Immunodeficiency Virus; IREC, Institutional Research Ethics committee; MTRH, Moi Teaching and Referral Hospital; NACADA, National Authority for the Campaign Against Alcohol and drug Abuse; PHQ-9, Patient Health Questionnaire-9; RCT, Randomized Controlled Trial; SMD, Standardized Mean Difference; SSA, Sub-Saharan Africa; SSBI, Single Session Brief Intervention; SUD, Substance use disorder; TIDieR, Template for intervention description and replication; US, United States; WHODAS, World Health Organization Disability Assessment Schedule; WHO-QOL BREF, World Health Organization-Quality of Life Brief Version.

Version [WHO-QOL BREF]) scores between baseline and month 3; (ii) Fidelity to the intervention assessed using fidelity checklists.

## Results

This pilot met most of the predefined minimum requirements for feasibility. For instance, 96.9% of those meeting eligibility criteria consented to participate (benchmark was 80%), and 100% of those who consented were willing to be randomized to either study arm. Youth reported that young people who use substances can be most effectively recruited from community settings. The SSBI showed a small effect on reducing total ASSIST-Y (Standardized Mean Difference [SMD] −0.33 95% Confidence Interval [CI] −0.83,0.16) scores in the intervention group compared to the control. There was a moderate improvement in the quality of life for the intervention group compared to the control (SMD −0.41 CI −0.91,0.09). The intervention had no effect on depression (SMD 0.23 CI −0.27,0.72) and anxiety symptoms (SMD 0.70 CI 0.19,1.2) at month 3.

## Conclusion

It is feasible to conduct a randomized controlled trial of a peer-delivered SSBI for youth with moderate risk substance use in Kenya.

## Trial registration

ClinicalTrials.gov NCT05545904 Registration date: 16/09/2022, https://clinicaltrials.gov/study/NCT05545904.

---

## 1. Introduction

In 2012, the United Nations set an ambitious goal to ensure healthy lives and promote the well-being for all ages by 2030 [1]. Achieving this goal will require that effective interventions to address substance use are implemented. This is particularly true for young people for whom substance use is a major impediment to their health and social well-being [2–4]. Substance use among youth contributes to risky sexual behavior [4,5], poor educational outcomes [3] increased risk of HIV and sexually transmitted infections (STIs) [6,7], poor mental health [5,8] and significant disability [2]. In the 2010, Global Burden of Disease (GBD) study, substance use disorders (SUDs) were the second leading cause of disability among the mental and SUDs for children and adolescents aged 0–24 years [2]. In that study, 7,173,000 Disability Adjusted Life Years (DALYS) were attributable to SUDs representing 25% of total DALYs from mental and SUDs in that age group [2].

Youth in sub-Saharan Africa (SSA) are particularly vulnerable to the effects of substance use because of poor substance use control policies [9] and limited opportunities for treatment [10–13]. Brief interventions (BIs) for substance use are recommended as a public health strategy for prevention and early intervention for

addressing youth substance use problems [14,15]. BIs typically comprise of screening to determine the level of substance involvement, followed by counseling based on motivational interviewing principles [16]. The intervention typically lasts 1–4 sessions [14,17].

Several studies, predominantly from high-income settings, have examined the effectiveness of BIs for youth substance use, with some demonstrating evidence of effectiveness. For instance, in the US, Winters et al [18] in a randomized controlled trial (RCT), evaluated the effectiveness of a two-session BI for youth in a school setting. The authors found that at 6 months, the BI resulted in a significant reduction in cannabis and alcohol use compared to the control condition. In an emergency department–based study conducted in the US, a 20-minute single-session BI, supplemented with a telephone booster, resulted in reduced cannabis use among youth at 12 months compared with the control group [19]. Meta-analytic findings are mixed. Steele et al found that BIs based on motivational interviewing compared to treatment as usual resulted in reduced alcohol but not cannabis use among adolescent's aged 12–20 years [20]. Carney and Myers [21] reported small but significant BI effects on youth substance use, while Samson and McHugh [21] found no BI effect for alcohol use among adolescents and young adults in emergency departments.[22]

Little research has evaluated the effectiveness of BIs for youth in Africa. We recently conducted a scoping review to map the substance use BI on the continent and identified very limited evidence focused on youth [23]. One study evaluating preliminary effectiveness of a two-session BI using a quasi-experimental approach reported reductions in cannabis and alcohol use at one month post intervention among South African adolescents aged 13–17 years [24].

Overall, evidence on the effectiveness of BIs for youth substance use remains inconclusive, and data from Africa are particularly limited, underscoring the need for further research in this area.

Over the past few years, the role and importance of peer providers in the delivery of mental health interventions for youth has gained traction in both low [25,26] and high-income countries [27,28]. Research findings across the globe support feasibility of youth peer-led interventions for depression and anxiety [27], building resilience [25], substance use [26,28], and for addressing bullying [29]. Peer providers are often age-mates or near age-mates of the youth and are well positioned to offer relatable and empathic support [30]. Secondly, peer-delivery is an important strategy for delivering youth mental health interventions in SSA because of shortages of the mental health workforce and large treatment gap in the region [31].

Globally, few studies have explored the effectiveness of peer-delivered substance use BIs. In the US, Bernstein et al [19], in a pilot RCT explored the effects of peer-delivered BI (one face to face session plus telephone booster) for youth and adolescents and found that the intervention resulted in a significant reduction in cannabis use at 12 months compared to the control. The peer providers in that study were aged 25 years and below. In a recent scoping review we conducted, we found no studies from Africa that examined the effectiveness of peer-delivered BIs for adolescents and youth in the region [23].

To fill this research gap, in 2021, we adapted a single-session substance use BI (SSBI) for peer-delivery and for youth in western Kenya [32,33]. The SSBI was based on the World Health Organization (WHO) Alcohol, Smoking & Substance Use Involvement Screening Test (ASSIST)-linked BI, which is grounded in the Feedback Responsibility Advice Menu of options Empathy Self-efficacy (FRAMES) model and motivational interviewing principles [16]. We then conducted an open trial and applied mixed methods to explore feasibility and acceptability of the SSBI from the perspective of youth, peer-providers and healthcare workers [32]. Quantitative data was collected using the dissemination and implementation science measures[33,34], and qualitative data collection was guided by the Consolidated Framework for Implementation Research (CFIR) [35]. Findings supported acceptability and feasibility of the SSBI with recommendations for improvement [32]. In 2022 we received additional funding to conduct a pilot feasibility RCT of the SSBI and to further explore intervention acceptability from the perspective of youth in preparation for a large-scale trial. The aim of this paper is to report on the pilot feasibility and preliminary effectiveness outcomes of that study. Findings on SSBI acceptability explored guided by Sekhon's theoretical framework of acceptability [36,37] have been published elsewhere [37].

## 2. Materials and methods

### 2.1. Trial design

This was a mixed methods study. To test the feasibility of a future definitive trial, we used a single-blind, parallel, pilot RCT study design. Participants were randomly allocated in a 1:1 ratio to either the SSBI or the control condition. Qualitative methods were used to explore youths' views on strategies for recruiting participants in a future large-scale trial.

**Adjustments made to the trial after commencement:** Because of practical challenges we faced during project implementation, we made three key adjustments to the protocol after commencement. These are summarized below:

(i) The initial plan was to recruit all participants then randomize to intervention or control. This led to significant dropout rates. We then decided to randomize at the point of recruitment. The details of both approaches have been described below.

(ii) Secondly, because we faced challenges with recruitment during this pilot, we amended the protocol and conducted three focus group discussions (FGDs) with 15 youths to obtain their perceptions on potential recruitment strategies for a future full-scale trial.

(iii) During the three-month follow-up, some youth had moved out of the study area and were unreachable. We therefore amended the protocol to include conducting the 3-month assessments online via Zoom.

### 2.2. Setting

We recruited participants from the adolescent HIV clinic (Rafiki clinic) run by the Academic Model Providing Access to Health Care-Kenya (AMPATH-Kenya) [38]. AMPATH-Kenya is a partnership between Moi Teaching and Referral Hospital (MTRH), Moi University, and a consortium of North American Universities that supports a large HIV care program along with other care, research, and education programs. Rafiki clinic has a total enrollment of 800 adolescents (aged 15–24 years), offers mainly HIV care services, but also pre-exposure prophylaxis, contraceptive care, counseling, and recreational activities, e.g., salsa dance. Eighty percent of adolescents attending the clinic are HIV positive. Eight peer providers aged 18–26 years, work full-time at Rafiki.

### 2.3. Study participants

Eligible participants were youth presenting for care at the Rafiki clinic, aged 15–24 years, and who had moderate risk substance use as defined by the Alcohol, Smoking & Substance Use Involvement Screening Test for Youth questionnaire (ASSIST-Y) scores [7]. We excluded youth who declined to assent/consent, and youth with high-risk substance use.

### 2.4. Sample size

We aimed to have 25 participants per group (Total N = 50). This sample size was considered large enough to inform us about the practicalities of delivering the SSBI and the control and was in line with existing pilot RCT recommendations for proportionality to planned definitive RCT [39] (i.e., we had ≥ 90% power to detect a small, standardized effect size of 0.2 [an effective SSBI will result in a 20% difference in the reduction in the total ASSIST-Y scores between SSBI and psycho-education arms]) [41]. The small, standardized effect size is consistent with meta-analytic evidence from studies evaluating the effectiveness of BIs among adolescents and youth, which demonstrates 10–20% reductions in substance use [21,40].

Because we anticipated 20% attrition between recruitment and baseline data collection, and a further 20% attrition between baseline data collection and the 3-month data collection, we targeted to recruit 36 participants per arm (total 72 participants). The attrition rates were based on a prior study testing a substance use intervention in our setting [41,42].

While the primary purpose of sample size calculation was not to make statistical inferences based on p-values, we conducted a power analysis to facilitate our interpretation of findings from our exploratory analyses. This sample size is therefore appropriate for feasibility assessment, and the study is not sufficiently powered to detect differences in clinical outcomes.

## 2.5. Participant recruitment

We aimed to recruit 36 males (18 aged 15–17 years and 18 aged 18–24 years) and 36 females (18 aged 15–17 years and 18 aged 18–24 years), with equal allocation to the SSBI and psychoeducation arms.

A trained research assistant approached all youth presenting for care. Youth were first screened using ASSIST-Y to determine eligibility. Only those with moderate risk substance use were eligible. Moderate risk substance use meant that all substances endorsed by the youth were at moderate risk level. Any youths with at least one high risk substance use were therefore excluded and were referred to the alcohol and drug abuse clinic at MTRH. Scores corresponding to moderate risk substance use on the ASSIST-Y are as follows: tobacco products [2–11], alcohol [5–17], cannabis [2–11], cocaine [2–8], amphetamine-type stimulants [2–8], sedatives [2–6], hallucinogens [2–8], inhalants [2–8], opioids [2–6] and 'other' drugs [2–6] (See Supporting File 1 for a complete description of ASSIST-Y Scoring). Eligible youth were then explained the study procedures and assent/consent sought in either English or Swahili. For youth aged 15–17 years, parental/guardian consent was obtained in addition to youth assent. Consenting/assenting was done in a private room within the clinic.

**2.5.1. Adaptations to youth recruitment processes.** In response to challenges related to study procedures, youth recruitment was implemented in two phases – an initial recruitment per protocol (phase 1), and an adapted recruitment process (phase 2). The approach taken during phase one was to recruit all 72 participants first, then randomize to either treatment or control. During phase 1, a total of 332 youths were approached. All of them agreed to be screened for eligibility. Of the 332, 247 were excluded because they had never used substances, and 12 had high risk use. One female aged 18–24 years with moderate risk use declined to consent and reported that she was not comfortable participating in the study. The parents of two youths aged 15–17-year-olds declined to consent and to give reasons for the refusal. In the end, we recruited 70 youths with moderate risk use for whom we had either consent or assent, who were then randomized to treatment vs control (n = 35 per arm).

During recruitment, we struggled to reach the minimum sample size for the four participant groups. It was particularly difficult to recruit youths (both male and female) aged 15–17 years. In the end we recruited more males than females and more 18–24 year olds than 15–17-year-olds (Table 1). Phase 1 of recruitment occurred between April 17 and June 13, 2023.

We then called the youths and asked them to return for baseline data collection and delivery of the intervention or control in July 2023. Forty-seven youth returned for baseline data collection resulting in a 65.3% attrition rate (Table 1). Of the 23 youth who did not return for baseline data collection, 19 were unreachable on phone and four were out of town. The average time between recruitment and baseline data collection was 84 days. Table 1 below shows the number of youth participants recruited during phase 1 by gender and age, and attrition between recruitment and baseline assessment.

Because we did not meet the minimum sample size, we embarked on a second phase of recruitment. During phase 2 we agreed to randomize and intervene at the point of recruitment to prevent attrition. The goal of phase two was to recruit 23 participants. Because it was challenging to recruit 15–17-year-olds, we agreed to recruit male and female 18–24-year-olds (12 male, 11 female) only during phase two. We also agreed not to replace the two youth (aged 15–17 years; one female and one male), whose parents had declined to consent because it had been challenging to recruit 15–17-year-olds.

During phase two, a total of 101 youths aged 18–24 years were approached. All of them agreed to be screened for eligibility. Of the 101, 67 were excluded because they had never used substances, and 11 had high risk use. In the end we had 23 who had moderate risk use from whom we had consent. Of the 23, 12 were assigned to treatment and 11 to

**Table 1. Phase 1 recruitment: Number of youth participants recruited by gender and age and attrition between recruitment and baseline assessment.**

| | Number (No.) recruited[a] | | Total recruited | No. returning for base-line assessments | Attrition rate (No. returning for baseline assessments/ Total recruited X 100) |
|---|---|---|---|---|---|
| | 15-17 years | 18-24 years | | | |
| **Male** | 16 | 28 | 44 | 32 | 72.7%[b] |
| **Female** | 8 | 18 | 26 | 15 | 57.7%[c] |
| **Total recruited** | **24** | **46** | **70** | | |
| **No. returning for baseline assessments** | 20 | 27 | 47 [f] | | |
| **Attrition rate (No. returning for baseline assessments/ Total recruited X 100)** | 83.3%[d] | 58.7%[e] | 65.3%[g] | | |

[a] Although stratified randomization by age and sex was planned (four groups each with 18 youth), balance between groups was not achieved; [b] Attrition rate among male participants; [c] Attrition rate among female participants.

[d] Attrition rate among 15–17-year-olds; [e] Attrition rate among 18–24-year-olds; [f] The required sample size was not achieved during Phase 1 of recruitment; [g] Overall attrition rate between enrollment and baseline data collection during Phase 1 of recruitment.

control. Baseline data was collected, and implementation of the intervention was done for these 23 at the time of recruitment. Phase two recruitment occurred between 4th and 22nd August 2023.

Overall, during both phases, we approached 433 youth. Of these 340 were excluded (314 because they had never used substances, 23 because they had high risk use for at least one substance, and three declined to consent [one youth after screening positive for moderate risk substance use, declined to consent saying that she was not comfortable participating in the study; the parents of two minors who had moderate risk substance use declined to consent]. Ninety-three were randomized: SSBI (n = 47) and control (n = 46). Of these, 70 underwent baseline assessments (23 did not return for baseline assessments during phase one of recruitment) and received either the SSBI or psychoeducation (Fig 1).

**Block sizes achieved by stratum:** Despite the use of stratified permuted block randomization to promote between-group balance by sex and age, this balance was not fully achieved. Table 2 summarizes the numbers of participants enrolled within each stratum for the SSBI and psychoeducation arms. For each study arm, blocks were designed to include a planned stratum size of nine participants. However, the planned stratum sizes were not met for male and female participants aged 15–17 years (Table 2).

## 2.6. Intervention and control

**SSBI (Intervention):** A description of the SSBI is provided in this section guided by the 'template for intervention description and replication' (TIDieR) whose goal is to ensure comprehensive intervention reporting to facilitate replication and further research [42]. The template recommends describing interventions by specifying the name, underlying theory and rationale, materials used, delivery procedures, provider characteristics, mode and setting of delivery, dose, duration and intensity, any adaptations or modifications, and fidelity assessment methods and findings [42].

Participants randomized to the intervention arm underwent a SSBI delivered by a trained peer-provider. The SSBI was focused on the highest scoring substance, or if two substances scored the same, the one identified as most problematic by the youth. The goal of the SSBI is to encourage youth with moderate risk use to cut down or stop using substances altogether [16]. The SSBI is based on motivational interviewing principles (i.e., open-ended questions, affirmation, reflections, summarizing), and the FRAMES model. The latter consists of: providing **feedback** on screening results; ensuring **responsibility** on the part of the youth; giving clear **advice** to stop/cut down on substance use; discussing a **menu of options** (alternative healthy behaviors to engage in); expressing **empathy** and encouraging **self**-efficacy [16]. The SSBI was conducted in 11 steps as follows: 1. Asking youth if they are interested in seeing their ASSIST-Y scores; 2. Providing

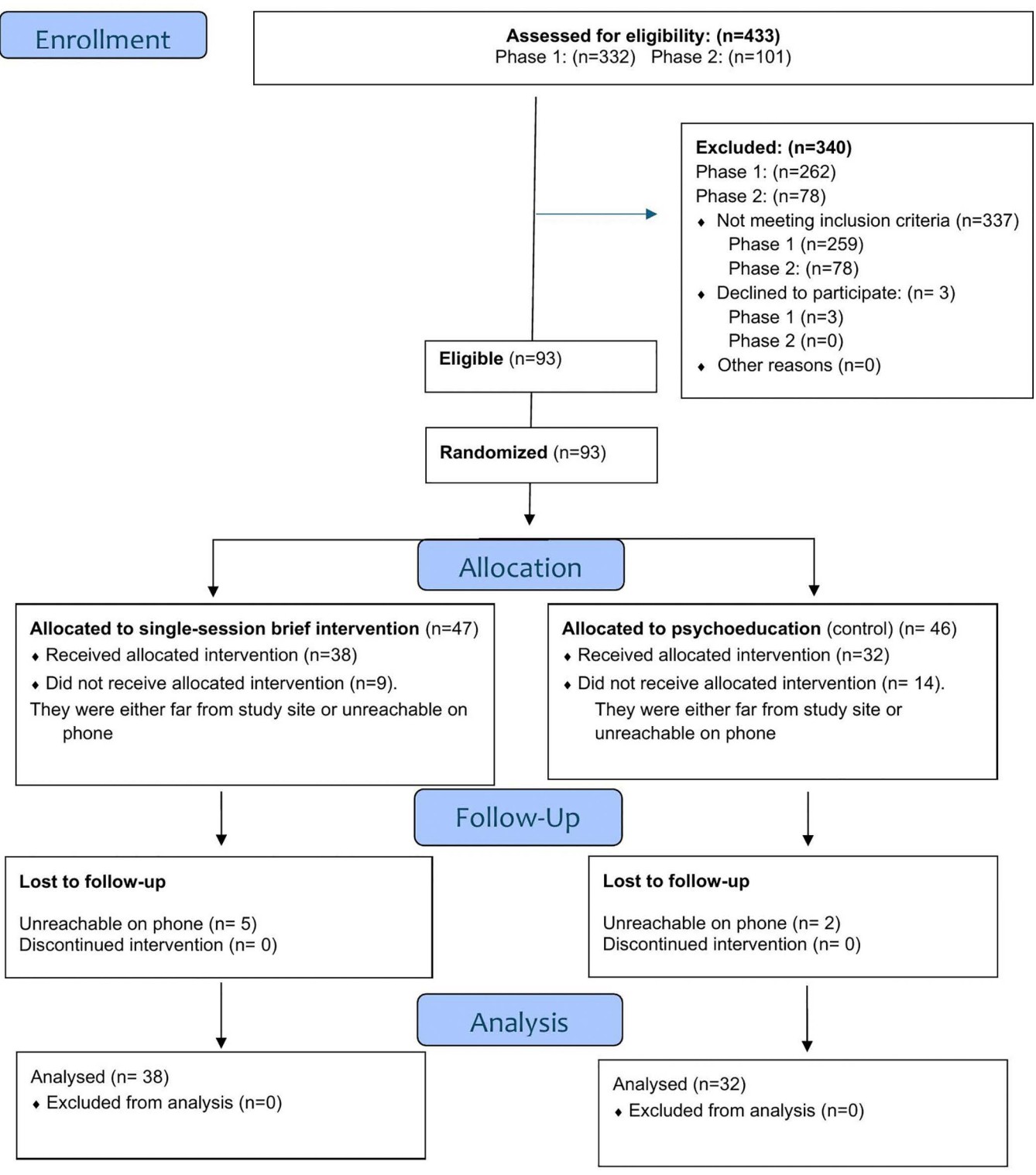

**Fig 1. Consort flow chart.**

**Table 2. Block sizes achieved by stratum.**

|  | Single-Session Brief Intervention | | Psychoeducation | |
| --- | --- | --- | --- | --- |
|  | **Male** | **Female** | **Male** | **Female** |
| **15-17 years** | 7 | 3 | 5 | 2 |
| **18-24 years** | 17 | 11 | 16 | 9 |

personalized feedback to youth about their ASSIST-Y scores; 3. Giving youth advice about how to reduce risk associated with substance use; 4. Allowing youth to take ultimate responsibility for their choices; 5. Asking youth how concerned they are by their ASSIST-Y scores; 6. Weighing up the good things about using the substance against the; 7. less good things about using the substance; 8. Summarizing and reflecting on youth' statements about their substance use with emphasis on the 'less good things'; 9. Asking youth how concerned they are by the 'less good things'; 10. An assessment of readiness or confidence to initiate change using the readiness steps; 11. Giving youth take-home materials to bolster the SSBI, i.e., the ASSIST-Y feedback report card and menu of options [16].

A guide for this intervention was adapted from the WHO ASSIST-linked BI manual [16] as part of an earlier pilot project and was translated to Swahili for this project [33]. One peer provider working full-time at Rafiki (female, aged 24 years) delivered the program face to face in individual sessions. The intervention was delivered in a private space within Rafiki clinic. The peer provider welcomed the youth to the session, introduced herself and the goal of the session, then conducted the intervention.

The peer provider had received a 5-day training on the SSBI as part of a previous pilot project [32]. For the current study, the peer provider underwent a two-day refresher training. The refresher training was conducted by psychologists J.B. and G.K, both of whom have extensive training and experience in conducting substance use and mental health interventions with adolescents. During the refresher training, the peer provider practiced counseling skills, including reflection and summarizing, as well as the SSBI steps, through role-plays conducted under the guidance of the trainers.

During the first week of SSBI delivery, F.J., J.B., and G.A. held daily meetings with the peer provider to check on any challenges. Thereafter weekly supervision meetings were held with the peer provider to offer continual training in skills as needed, and to ensure fidelity to the SSBI. All SSBI sessions were recorded to facilitate fidelity assessments. On average, the SSBI sessions lasted 9 minutes 55s. Fidelity to the SSBI was assessed and this process and findings have been described in this paper (See Section 2.9.2 for a description of the process of conducting fidelity assessments). Table 3 provides a summary of the SSBI guided by the TiDiER template.

**Control:** Participants assigned to the control arm participated in a substance use education intervention. This intervention entailed review of material in the National Authority for Campaign Against Alcohol and Drug Abuse (NACADA) substance use education manual for adolescents [43]. The manual contains summarized and simple information on the harms and myths related to alcohol, tobacco, cannabis, prescription medication, and khat use, substances [43]. Each youth in the control arm underwent psychoeducation on the highest scoring substance or the most problematic one. The psychoeducation session was followed by a question-and-answer session. This education intervention was delivered over a single session by the same peer-provider who implemented the SSBI. On average, the psychoeducation sessions lasted 7 minutes.

## 2.7. Study outcomes

**2.7.1. Primary outcome.** The primary outcome for this study included feasibility of a future definitive trial evaluating the effectiveness of the SSBI. To test the feasibility of conducting a future definitive trial, we examined the study participation rate, proportion of participants meeting inclusion criteria who get excluded, proportion of participants willing to be randomized, study completion rate, participant burden, and data completeness. The predefined feasibility benchmarks are described in Table 4 below.

**Table 3. Summary of SSBI guided by TiDiER template.**

| TIDieR component | Intervention description |
|---|---|
| **Name of the intervention** | Single-session brief intervention (SSBI) |
| **Theory and rationale for the intervention** | SSBI is based on motivational interviewing principles and the FRAMES model[a] [16]<br>The goal of both models is to foster intrinsic motivation among youth to reduce or discontinue substance use [16]. |
| **Physical or informational material provided to participants or used in intervention delivery** | ASSIST-Y feedback report card and menu of options. |
| **Processes/procedures carried out in delivering the intervention;** | The intervention involves the peer-provider asking the youth about their interest in viewing their ASSIST-Y scores; providing personalized feedback on the scores; offering risk-reduction advice; emphasizing youth autonomy; exploring concern about scores; weighing perceived pros and cons of substance use; summarizing and reflecting statements with emphasis on cons; assessing concern about these cons; evaluating readiness to change; and providing take-home materials (ASSIST-Y feedback report card and menu of options) to reinforce the SSBI. |
| **Details of provider** | Female, aged 24 years; trained to deliver the SSBI over a 5-day training in earlier pilot [32] and 2-day refresher training provided for current pilot. |
| **Mode of delivery;** | Face-to-face |
| **Where the intervention was delivered;** | Adolescent Clinic at a Tertiary level health facility in Kenya (Rafiki clinic, MTRH) |
| **Intervention dose, duration, intensity;** | Single session intervention; duration (average time of delivery 9 minutes 55 seconds) |
| **Adaptations made; and modifications during the study;** | Translated to Swahili (see Section 2.10 on translation/adaptation procedures) |
| **Fidelity assessments and findings** | Fidelity to the SSBI was assessed by audio-recording of all sessions (n = 70) and rating them using a checklist of key elements of the SSBI (Section 2.9.2)<br>The mean fidelity scores were 33.82 out of 34 for the SSBI (Section 3.6) |

[a]Feedback, Responsibility, Advice, Menu of options, Empathy, Self-efficacy.

**Table 4. Predefined feasibility benchmarks.**

| Measure | Definition | Administration time point | Predefined benchmarks to establish feasibility for conducting a full-scale randomized trial |
|---|---|---|---|
| Study Participation Rate | Number of participants who consent to take part in the study divided by the number of eligible patients. | Throughout recruitment | 80% of those who meet eligibility criteria consent to participate |
| Proportion of participants meeting inclusion criteria who get excluded | Number of participants excluded divided by number meeting inclusion criteria. | Throughout recruitment | 80% of those meeting inclusion criteria are not excluded |
| Proportion of participants willing to be randomized | Number of participants consenting to participate divided by number willing to be randomized to either study arm. | During recruitment | 80% of those who consent are willing to be randomized to either study arm |
| Study Completion Rate | Number of participants who complete the month 3 assessments divided by the number of participants enrolled in each study arm | Baseline, month 3 | 80% complete both month 3 assessments in each study arm |
| Participant Burden | Time required to complete data collection at each assessment time point. | Baseline | 80% of participants complete study assessments and the SSBI in less than 90 minutes at baseline |

**2.7.2. Secondary outcomes.** Secondary outcomes for this study included preliminary SSBI effectiveness, and fidelity to the intervention.

Preliminary effectiveness of the SSBI was assessed as follows:

• change in substance use scores (measured using ASSIST-Y) from baseline to three months post-intervention

- change in quality-of-life scores (measured using the Brief Version WHO-Quality of life tool) from baseline to three months post-intervention.

- change in depression (PHQ-9) scores from baseline to three months post-intervention

- change in generalized anxiety disorder (GAD-7) scores from baseline to three months post-intervention.

Intervention fidelity was assessed using a checklist of key elements of the SSBI. The checklist has been provided in Supporting File 2.

Reporting for this study is guided by the CONSORT checklist for pilot and feasibility trials[44] (S3 File).

## 2.8. Randomization

Following enrollment, participants were randomized to one of the two study groups (allocation ratio: 1:1), SSBI or psychoeducation. Stratified permuted block randomization procedures was used to achieve between-group balance by sex and age. We were however unable to achieve between group sex and age balance (Table 4). Randomization lists were prepared by the study biostatistician and uploaded into a REDCap database to allow for concealed allocation among the study team while maintaining a blinded statistical team.

## 2.9. Data collection and measures

**2.9.1. Baseline data collection.** At baseline the trained research assistant collected data on socio-demographics and mental health characteristics as follows.

**A researcher designed questionnaire** was used to collect socio-demographic data (age, sex, parental status, living arrangement, level of education) at baseline. The research assistant then administered the socio-demographic questionnaire, Generalized Anxiety Disorder-7 (GAD-7), Primary Health Questionnaire (PHQ-9), and the World Health Organization-Quality of Life Brief Version (WHO-QOL BREF) to consenting youth.

**Patient Health Questionnaire-9 (PHQ-9)** [45] was used to collect data on depression at baseline. The PHQ-9 is a valid and reliable tool for measuring severity of major depression [45] and has been validated among Kenyan adolescents [46].

The **Generalized Anxiety Disorder-7 (GAD-7) scale** [47] was used to collect data on Generalized Anxiety Disorder (GAD) at baseline. GAD-7 is a valid and reliable tool for measuring severity of GAD [47]. Osborn et al [46] examined the psychometric properties of the GAD-7 among Kenyan adolescents and reported that the reliability was adequate.

The **Brief Version of the WHO-Quality of Life [WHO-QOL BREF]** tool [48] was used to assess participant quality of life at baseline and month 3. The tool is comprised of 26 questions organized into four domains: social relationships, environment, physical health, and psychological [48]. The WHO-QOL BREF has been validated for use among adolescent populations [49] but not specifically in Kenya.

**ASSIST –Y questionnaire:** ASSIST is a valid and reliable tool that asks about frequency of substance use in the past 3 months [16]. The adolescents' version of the tool (ASSIST-Y) was developed by experts from Adelaide University in Australia [50]. The ASSIST-Y enquires about lifetime use of nine substances, i.e., tobacco products, alcohol, cannabis, amphetamines, cocaine, inhalants, sedatives, opioids and hallucinogens. Endorsement of lifetime use is followed by an assessment of substance use in the past 3 months [50]. The level of substance involvement is categorized as moderate, or high risk and cut-off scores vary for each substance [50] (Supporting File 1). Unlike the adult version, the ASSIST-Y has no 'low risk' category [50] (Supporting File 1). The ASSIST has been validated among adolescents in SSA [51] and has been used in Kenya among youth [32]. The ASSIST-Y was adapted by our team in an earlier pilot to include local names and varieties of substances [33]. Of note is that for amphetamines we enquired about Khat use. Khat is a plant cultivated in Eastern Africa whose leaves are chewed for their stimulant effects. Khat leaves contain stimulants such as cathinone, cathine and norephedrine [52].

**2.9.2. Fidelity assessments.** Fidelity to the SSBI was assessed by audio-recording all sessions (n = 70) and rating them using a checklist of key elements of the SSBI. Five of the recordings were rated by F.J. and J.B and a percentage agreement of 96.5% achieved. We discussed items that had been rated differently and built consensus on how to score. Thereafter, the remainder of the recordings were independently rated by F.J. and J.B.

The psychoeducation sessions were rated using the SSBI fidelity checklist to ensure that the sessions had minimal SSBI related content. Five recordings were rated by F.J. and J.B., and a percent agreement of 94.1%. Thereafter, the remainder of the recordings were independently rated by F.J. and J.B.

**2.9.3. Data collection at month 3 (follow-up data collection).** At month three (between October and November 2023), the youth were invited for a repeat assessment. Data was collected from each participant using the ASSIST-Y, WHO-QOL BREF, PHQ-9, and GAD-7. Some youths were not available for assessments physically because they had travelled out of town. We therefore amended the protocol and conducted four assessments via the Zoom online meeting platform (https://zoom.us/). A three-month follow-up period was selected because per Diagnostic and Statistical Manual fifth edition (DSM-5) [53] a person with a SUD is in early remission if they do not meet criteria for SUD for three continuous months. At month three, the study completion rate was 33/47 (86.8%) in the SSBI arm, and 30/46 (93.8%) in the psychoeducation arm (Table 5).

**2.9.4. Qualitative data collection.** At month three, we conducted three FGDs with 15 youth to explore perceptions of optimal strategies for recruiting participants in a future definitive RCT. The FGDs were conducted by two experienced facilitators independent of the research team and were held on 4, 5, and 12 December 2023, with each session lasting an average of 1 hour and 45 minutes.

## 2.10. Translation/adaptation of study material and the SSBI manual

All study materials including the quantitative tools, FGD interview guides, the SSBI manual, and the consent and assent forms were translated to Swahili. The need to translate the SSBI into Swahili was identified as a recommendation from the youth in a prior pilot study [32]. The tools were first translated to Swahili by experienced translators with a good command of both the English and Swahili languages. These were then back translated to English by a separate set of translators. A team comprising translators, back-translators, and mental health experts discussed the translated documents and resolved any differences to arrive at the final Swahili versions. The translation process was guided by the WHO-Disability Assessment Schedule (WHO-DAS) 2.0 translation protocol [54].

## 2.11. Data management and confidentiality

Study data was recorded onto standardized paper forms at the time of collection. Data was anonymized by assigning each participant an unidentifiable study ID number at the time of enrolment, which was used to identify them for all study materials. Paper data forms were immediately filed and stored in a locked cabinet. Signed study consent forms were filed and stored separately from data forms to maintain participant anonymity. Study data was subsequently entered into a secure REDCap database by a research assistant. Audio recordings of the study sessions were transferred onto a secure online platform (Microsoft SharePoint) and deleted from their original recording device at the time of transfer. To ensure completeness, we conducted double data entry and resolved discrepancies.

**Table 5. Study completion at month 3.**

| Allocation | Baseline assessments | 3-month assessments | Study completion rate |
|---|---|---|---|
| Intervention (SSBI) | 38 | 33 | 86.8% |
| Control (Psychoeducation) | 32 | 30 | 93.8% |
| Total | 70 | 63 | 90% |

### 2.12. Data analysis

**2.12.1. Quantitative data analysis.** Descriptive statistics were used to summarize socio-demographic and mental health characteristics of the youth. Means and standard deviations were used to summarize continuous normally distributed data, medians and ranges for continuous non-normally distributed data, and frequencies and percentages for categorical data. The proportion of patients meeting each of the feasibility endpoints (eligibility, recruitment, and attrition rates), with accompanying 95% confidence intervals (CIs), was calculated. Levels of fidelity were obtained by calculating mean scores across all items of the fidelity checklist.

To investigate the preliminary effectiveness of the SSBI on substance use, depression, anxiety, and quality of life, we calculated the standardized mean difference (SMD), also known as Cohen's d [55]. This was calculated as (mean of control group – mean of treatment group)/ the pooled standard deviation. Additionally, we calculated the corresponding 95% CI for each SMD, which is important in pilot studies to provide a range for the effect size, adding interpretive context on variability and the precision of the effect estimate without implying statistical significance [55]. SMD is widely used in pilot studies to assess group differences in a descriptive manner, emphasizing effect magnitude rather than statistical inference [55,56]. It offers insight into practical differences without being influenced by small sample sizes, making it ideal for preliminary studies focused on feasibility and exploratory outcomes [56].

As noted by Fritz et al. [56], studies with different sample sizes but the same basic descriptive characteristics (e.g., distributions, means, standard deviations, CIs) will differ in their statistical significance values but not in their SMD estimates. For interpretation in our analysis, we followed Cohen's guidelines: The absolute value of the SMD indicates the strength of the effect, with larger values reflecting stronger effects. Therefore, an SMD of 0.2 indicates a small effect, 0.5 a moderate effect, and 0.8 or higher a large effect [55]. In this context, SMDs near zero suggest balanced characteristics across groups, while larger values reflect practical differences, enhancing understanding of the SSBI's potential impact and guiding future studies [57].

**2.12.2. Qualitative data analysis.** The audio-recorded Interviews were transcribed verbatim then entered into Dedoose for analysis. The transcripts were reviewed, and initial coding was done separately by the two facilitators who conducted the FGDs using an inductive approach. The two discussed the codes and sub-codes and resolved initial disagreements to develop a refined codebook (See Supporting File 4 for codebook). The final coding of the transcripts was done by both facilitators using the refined codebook.

F.J., and the two facilitators then performed a thematic analysis to identify codes that fit into themes that addressed the key goal of the FGDs which was to understand the best approach to recruit youth who use substances in a future trial. The themes were developed and defined through a process of discussion amongst the three (F.J., and the two facilitators) until a consensus was arrived at. The codes and sub-codes fit into four themes, i.e., best places to recruit youth who use substances; challenges to anticipate when recruiting youth who use substances and how to address those challenges; feasibility of snowballing as a recruitment strategy; ways to recontact youth for a follow-up session.

### 2.13. Ethics approval and consent to participate

Ethical approval to conduct the study was sought from the MTRH/ Moi University Institutional Research Ethics committee (IREC) and the Indiana University Institutional Review Board. All experimental protocols were approved by IREC (Approval Number IREC220/2022), and the Indiana University Institutional Review Board.

Prior to data collection, written informed assent was sought from the youth aged 15–17 years. In addition, written informed consent was obtained from parents of youth aged 15–17 years. Written informed consent was sought from youth aged 18–24 years. All methods and study procedures were carried out in accordance with the Declaration of Helsinki.

## 3. Results

### 3.1. Sociodemographic characteristics of the youth participants

A total of 70 participants were enrolled in the study. Most of them (75.7%) were aged between 18–24 years and were male (64.3%). Based on Cohen's [55] absolute value of the SMD, we found small differences in gender and age, and large differences in parental status between the SSBI and psychoeducation arms (Table 6).

### 3.2. Baseline mental health and substance use characteristics of the youth participants

#### 3.2.1. Lifetime substance use for the youth participants at baseline.
Among the participants, lifetime substance use was highest for alcohol (94.3%), followed by cannabis (39.1%), and tobacco (15.7%). There were small absolute SMD differences in the prevalence of lifetime substance use between the control and intervention arms at baseline (Table 7).

**Table 6. Socio-demographic characteristics of the youth participants.**

| Characteristic | Study Arm | | | Difference[3] | 95% CI[34] |
|---|---|---|---|---|---|
| | Overall, N = 70[1] | Psychoeducation N = 32[2] | SSBI N = 38[2] | | |
| **Age** | 19.7 (2.6) | 20.0 (2.5) | 19.4 (2.7) | 0.23 | −0.24, 0.71 |
| **Age Group** | | | | 0.10 | −0.37, 0.57 |
| *Old (18–24)* | 53.0 (75.7%) | 25.0 (78.1%) | 28.0 (73.7%) | | |
| *Young (15–17)* | 17.0 (24.3%) | 7.0 (21.9%) | 10.0 (26.3%) | | |
| **Gender** | | | | 0.05 | −0.42, 0.52 |
| *Female* | 25.0 (35.7%) | 11.0 (34.4%) | 14.0 (36.8%) | | |
| *Male* | 45.0 (64.3%) | 21.0 (65.6%) | 24.0 (63.2%) | | |
| **Level of education** | | | | 0.50 | 0.02, 0.98 |
| *Complete primary* | 3.0 (4.3%) | 3.0 (9.4%) | 0.0 (0.0%) | | |
| *Complete secondary* | 23.0 (32.9%) | 10.0 (31.3%) | 13.0 (34.2%) | | |
| *Incomplete secondary* | 27.0 (38.6%) | 13.0 (40.6%) | 14.0 (36.8%) | | |
| *Tertiary +* | 17.0 (24.3%) | 6.0 (18.8%) | 11.0 (28.9%) | | |
| **Living arrangement** | | | | 0.19 | −0.28, 0.66 |
| *Alone* | 19.0 (27.1%) | 8.0 (25.0%) | 11.0 (28.9%) | | |
| *Family/relative* | 48.0 (68.6%) | 22.0 (68.8%) | 26.0 (68.4%) | | |
| *Friend/non-relative* | 3.0 (4.3%) | 2.0 (6.3%) | 1.0 (2.6%) | | |
| **Parental status** | | | | 0.60 | 0.12, 1.1 |
| *Both parents alive* | 33.0 (47.8%) | 12.0 (38.7%) | 21.0 (55.3%) | | |
| *Both parents died* | 15.0 (21.7%) | 7.0 (22.6%) | 8.0 (21.1%) | | |
| *One parent alive (Father)* | 9.0 (13.0%) | 3.0 (9.7%) | 6.0 (15.8%) | | |
| *One parent alive (Mother)* | 12.0 (17.4%) | 9.0 (29.0%) | 3.0 (7.9%) | | |
| *Missing* | 1 | 1 | 0 | | |
| **HIV status** | | | | 0.47 | −0.01, 0.95 |
| *Negative* | 11.0 (15.9%) | 3.0 (9.4%) | 8.0 (21.6%) | | |
| *Positive* | 46.0 (66.7%) | 25.0 (78.1%) | 21.0 (56.8%) | | |
| *Unknown* | 12.0 (17.4%) | 4.0 (12.5%) | 8.0 (21.6%) | | |
| *Missing* | 1 | 0 | 1 | | |

1 Mean (SD); n (%).

2 Mean (SD); % (n).

3 Standardized Mean Difference between psychoeducation and SSBI arms.

4 CI = Confidence Interval.

**Table 7. Lifetime substance use for youth participants at baseline.**

| Characteristic | Study Arm | | | Difference[2] | 95% CI[23] |
| --- | --- | --- | --- | --- | --- |
| | Overall, N = 70[1] | Psychoeducation N = 32[1] | SSBI N = 38[1] | | |
| **Tobacco products** | | | | 0.33 | −0.14, 0.80 |
| No | 59.0 (84.3%) | 29.0 (90.6%) | 30.0 (78.9%) | | |
| Yes | 11.0 (15.7%) | 3.0 (9.4%) | 8.0 (21.1%) | | |
| **Alcohol** | | | | 0.04 | −0.43, 0.51 |
| No | 4.0 (5.7%) | 2.0 (6.3%) | 2.0 (5.3%) | | |
| Yes | 66.0 (94.3%) | 30.0 (93.8%) | 36.0 (94.7%) | | |
| **Cannabis** | | | | 0.06 | −0.42, 0.53 |
| No | 42.0 (60.9%) | 19.0 (59.4%) | 23.0 (62.2%) | | |
| Yes | 27.0 (39.1%) | 13.0 (40.6%) | 14.0 (37.8%) | | |
| Missing | 1 | 0 | 1 | | |
| **Cocaine** | | | | 0.00 | −0.47, 0.47 |
| No | 70.0 (100.0%) | 32.0 (100.0%) | 38.0 (100.0%) | | |
| **Khat[4]** | | | | 0.45 | −0.04, 0.93 |
| No | 58.0 (85.3%) | 29.0 (93.5%) | 29.0 (78.4%) | | |
| Yes | 10.0 (14.7%) | 2.0 (6.5%) | 8.0 (21.6%) | | |
| Missing | 2 | 1 | 1 | | |
| **Inhalants** | | | | 0.23 | −0.24, 0.70 |
| No | 69.0 (98.6%) | 32.0 (100.0%) | 37.0 (97.4%) | | |
| Yes | 1.0 (1.4%) | 0.0 (0.0%) | 1.0 (2.6%) | | |
| **Sedatives or Sleeping pills** | | | | 0.17 | −0.30, 0.65 |
| No | 66.0 (95.7%) | 30.0 (93.8%) | 36.0 (97.3%) | | |
| Yes | 3.0 (4.3%) | 2.0 (6.3%) | 1.0 (2.7%) | | |
| Missing | 1 | 0 | 1 | | |
| **Opioids** | | | | 0.24 | −0.24, 0.71 |
| No | 68.0 (98.6%) | 32.0 (100.0%) | 36.0 (97.3%) | | |
| Yes | 1.0 (1.4%) | 0.0 (0.0%) | 1.0 (2.7%) | | |
| Missing | 1 | 0 | 1 | | |

1 n (%).

2 Standardized Mean Difference between Psychoeducation and SSBI arms.

3 CI = Confidence Interval.

**3.2.2. Mean ASSIST-Y scores for youth participants at baseline.** Table 8 presents the mean ASSIST-Y scores at baseline for both SSBI and psychoeducation groups across various substance use categories. The SMD and corresponding 95% CIs are reported for each factor. At baseline, there were small absolute SMD differences in substance specific ASSIST-Y scores, total ASSIST-Y scores and ASSIST-Y scores for the highest or most problematic substance between the SSBI and psychoeducation arms (Table 8).

Overall, there were no substantial differences between the psychoeducation and SSBI groups across any of the substance use categories at baseline, as indicated by the SMDs and CIs, all of which include zero. This suggests that both groups were well-balanced in terms of substance use characteristics prior to the intervention.

**3.2.3. Mean scores for PHQ-9, ASSIST-Y, WHO-QOL and GAD-7 for youth participants at baseline.** Table 9 presents the mean baseline scores for PHQ-9, ASSIST-Y, WHO-QOL, and GAD-7 for both the psychoeducation and SSBI groups, along with the SMD and 95% CIs. There were small SMD absolute value differences in depression scores

**Table 8. Mean ASSIST-Y scores for youth participants at baseline.**

| Characteristic | Study Arm | | | Difference[2] | 95% CI[23] |
| --- | --- | --- | --- | --- | --- |
| | Overall, N=70[1] | Psychoeducation N=32[1] | SSBI N=38[1] | | |
| Tobacco products | 0.7 (2.5) | 0.9 (2.9) | 0.6 (2.1) | 0.13 | −0.34, 0.60 |
| Alcohol | 5.7 (4.8) | 5.3 (4.3) | 6.1 (5.3) | −0.19 | −0.66, 0.29 |
| Cannabis | 1.8 (3.1) | 2.3 (3.5) | 1.4 (2.6) | 0.32 | −0.15, 0.79 |
| Cocaine | 0.1 (0.6) | 0.2 (0.9) | 0.0 (0.0) | 0.25 | −0.22, 0.73 |
| Khat | 0.5 (1.6) | 0.2 (1.2) | 0.7 (1.9) | −0.32 | −0.79, 0.16 |
| Sedatives or Sleeping pills | 0.1 (0.7) | 0.2 (0.9) | 0.1 (0.5) | 0.11 | −0.36, 0.58 |
| Opioids | 0.0 (0.0) | 0.0 (0.0) | 0.0 (0.0) | 0.00 | −0.47, 0.47 |
| Inhalants | 0.0 (0.0) | 0.0 (0.0) | 0.0 (0.0) | 0.00 | −0.47, 0.47 |
| Total ASSIST Y score | 8.9 (6.6) | 9.1 (6.8) | 8.8 (6.5) | 0.03 | −0.44, 0.50 |
| ASSIST-Y score for most problematic substance | 7.0 (4.3) | 6.7 (3.9) | 7.2 (4.7) | −0.10 | −0.57, 0.37 |

1 Mean (SD).

2 Standardized Mean Difference between Psychoeducation and SSBI arms.

3 CI = Confidence Interval.

**Table 9. Mean scores for PHQ-9, ASSIST-Y, WHO-QOL and GAD-7 for youth participants at baseline.**

| Characteristic | Study Arm | | | Difference[2] | 95% CI[23] |
| --- | --- | --- | --- | --- | --- |
| | Overall, N=70[1] | Psychoeducation N=32[1] | SSBI N=38[1] | | |
| Mean PHQ-9 scores | 5.3 (4.6) | 4.4 (4.0) | 6.0 (5.0) | −0.35 | −0.83, 0.12 |
| PHQ-9 Severity level | | | | 0.15 | −0.34, 0.63 |
| 1 - 4: Minimal depression | 34.0 (51.5%) | 16.0 (51.6%) | 18.0 (51.4%) | | |
| 10 - 14: Moderate depression | 7.0 (10.6%) | 3.0 (9.7%) | 4.0 (11.4%) | | |
| 15 - 19: Moderately severe depression | 3.0 (4.5%) | 1.0 (3.2%) | 2.0 (5.7%) | | |
| 5 - 9: Mild depression | 22.0 (33.3%) | 11.0 (35.5%) | 11.0 (31.4%) | | |
| Missing | 4 | 1 | 3 | | |
| Mean GAD-7 scores | 5.4 (4.5) | 3.9 (3.6) | 6.7 (4.9) | −0.67 | −1.2, −0.19 |
| GAD-7 severity level | | | | 0.60 | 0.12, 1.1 |
| Score 0–4: Minimal Anxiety. | 38.0 (55.1%) | 22.0 (68.8%) | 16.0 (43.2%) | | |
| Score 10–14: Moderate Anxiety. | 13.0 (18.8%) | 4.0 (12.5%) | 9.0 (24.3%) | | |
| Score 5–9: Mild Anxiety. | 16.0 (23.2%) | 6.0 (18.8%) | 10.0 (27.0%) | | |
| Score greater than 15: Severe Anxiety. | 2.0 (2.9%) | 0.0 (0.0%) | 2.0 (5.4%) | | |
| Missing | 1 | 0 | 1 | | |
| Mean WHO-QOL BREF score | 83.5 (12.9) | 86.8 (9.9) | 80.7 (14.5) | 0.50 | 0.02, 0.98 |

1 Mean (SD); n (%).

2 Standardized Mean Difference between Psychoeducation and SSBI arms.

3 CI = Confidence Interval.

between the psychoeducation and SSBI arms. There were lower GAD-7 scores in the SSBI arm compared to the psychoeducation arm and this difference was of medium effect size (Cohen's −0.67). Similarly, there was a moderate difference in WHO-QOL scores favoring the psychoeducation group (Cohen's 0.50). These differences indicate potential variations in baseline anxiety and quality of life between the two groups, with the SSBI group showing slightly higher anxiety and lower quality of life (Table 9).

## 3.3. Feasibility of a future definitive RCT

The study participation rate was 96.9%. Over two thirds (70.2%) of those assigned to the SSBI arm and 65.2% assigned to the psychoeducation arm, completed both baseline and three-month assessments. Table 10 provides complete data on the feasibility findings.

## 3.4. Strategies for recruiting youth in a future full-scale trial

### 3.4.1. Places to recruit girls aged 15–17 years.
Seven youths reported that girls this age who use substances could be found in schools. One said: *"[we can find them] in schools. You hold a conference that talks about drugs and how to avoid drugs and stuff, you just go and talk to all the students. You can give them some time where anybody can come and tell you about anything... When they close school they can come to [the study] …. and you can talk to them. (FGD 1, respondent 4).*

Six youths reported that the community or homes would be the best place to find girls aged 15–17 years who use substances. One youth reported: *"I can say at that age, 15 to 17, those guys are not exposed to the world, so most, you will [find them by going] to the village, maybe going house by house asking the children, because most of the children, you can't say are exposed as those who have finished school. So, they are not out here they are in the villages. So, you just … go to the villages and ask the kids questions. I know it may be hard they may not open up because of their parents and everything but try".* (FGD 1, respondent 3).

Four youths reported that this demographic could be found at parties. One reported that *"Most of them can be found where these things are like parties... So, if you organize something like a [party] and tell them that there are [substances] there, they will come, and that's where you will find them."* (FGD 2 respondent 2).

One youth each recommended hospitals and drug dens, and two youth recommended churches as good places to recruit youth.

**Table 10. Pilot feasibility findings.**

| Measure (pre-defined benchmark) | Findings |
|---|---|
| Study Participation Rate (80% of those who meet eligibility criteria consent to participate) | Phase 1 of recruitment: 70 youths consented to participate out of 73 who were eligible to participate. Phase 2 of recruitment: All 23 participants who were eligible to participate gave consent. Overall, 93 participants consented to participate in the study out of the 96 who were eligible resulting in a study participation rate of 96.9%. Feasibility benchmark achieved |
| Proportion of participants meeting inclusion criteria who get excluded (80% of those meeting inclusion criteria are not excluded) | Phase 1 of recruitment: 73 youth met inclusion criteria. Out of these, 26 were excluded from the study. Three did not give consent, and 23 did not return for baseline data collection because either they were far or were unreachable on phone. The proportion of participants meeting inclusion criteria who got excluded during phase 1 was therefore 26/73 = 35.6%. Phase 2 of recruitment, all 23 who were eligible to participate were included in the study. The proportion of participants meeting inclusion criteria who were excluded during phase 2 was therefore 0/23 = 0%. Overall, the proportion of participants meeting inclusion criteria who got excluded was 26/96 = 27.1%. In other words, 72.9% of those meeting inclusion criteria were not excluded. Feasibility benchmark not achieved |
| Proportion of participants willing to be randomized (80% of participants are willing to be randomized) | 100% were willing to be randomized for either study arm. Feasibility benchmark achieved |
| Study Completion Rate (80% complete month 3 assessments) | 86.8% assigned to the SSBI arm completed month 3 assessments, and 93.8% assigned to the psychoeducation arm completed month 3 assessments. Feasibility benchmark achieved |
| Participant Burden (80% of participants complete study assessments and the SSBI in less than 90 minutes at baseline) | At baseline, the mean time for study assessments was 15 min 51s and for SSBI was 9 min 55s. Altogether, baseline assessments and BI took an average of 25 min 46s. Feasibility benchmark achieved |

**3.4.2. Places to recruit boys aged 15–17 years.** Six youth reported that this demographic could be found at recreational and sporting activities such as football. One youth said: *"Mostly the boys play sports [like soccer]. You can find them there."* (FGD 3, *respondent* 2).

Five youths reported that boys aged 15–17 years could be found in schools. However, one youth disagreed with this:

*"School? No! School is... for girls. Girls are found with more drugs in schools than boys. So, school for boys, no* (FGD 1 respondent 2).

*School is very hard. Approaching a girl is easier than approaching a man. Men are so difficult when it comes to opening up. A man can't come and tell you that they are using drugs in school. That is very hard. Maybe a few like one percent. Men are difficult to approach, especially in school.* (FGD 1 respondent 1).

Three youths each recommended recruitment from the streets, and from gaming places.

*"let's just say in gaming hubs, at the PlayStation, betting places and stuff…"* (FGD 1, respondent 4).

One youth each recommended recruitment from the rescue center, and where substances are sold.
Four youths felt that hospitals were not a good place to recruit boys aged 15–17 years from. One said:

*"… no. Boys usually want to have fun more than girls. You know you can tell a girl that what she is using will spoil her life, but a boy can't [listen], no matter how much you convince them. I think the best place to find them is those gaming places… But for hospitals, they will say, "I go to hospital? How now. So that I can be told that they need to do what to me...?!"* (FGD 1, respondent 2).

Two youths, however, felt that it would be possible to recruit young people aged 15–17 years within youth dedicated clinics like Rafiki Clinic.

**3.4.3. Places to recruit girls 18–24 years.** Six youths reported that this demographic could be found at social events such as parties, concerts, religious events. One youth said: *"You can host a social event and introduce the topic during the event. In December we usually have the Jesus party. They like such things, and many young people are there. You can organize such a thing where there is a stage, and many young people are using [substances]. Something like a crusade."* (FGD 2, respondent 5).

Three youths reported that they can be found in the streets: *"Just those streets. Because they are of age, they don't fear anything they just [use substances] in public."* (FGD 2, respondent 3).

Three youths reported that they can be found in colleges: *"Most of them can …be found on campuses. At 18 to 24, they are big and financially stable…"* (FGD 2 respondent 1). Another said *"You can find them in campus events like campus night. Most of them show up. You know there is no one to control them there on campus, so they are free."* (FGD 2, respondent 2).

Two youths each reported that this demographic could be found in seminars and conferences for youth, and on social media; one youth each reported that they could be found in hospitals and where substances are sold.

**3.4.4. Places to recruit boys aged 18–24 years.** Four youths reported that boys aged 18–24 years who use substances could be found in gaming places (pool, movies, watching football).

*"[These boys can be found] playing pool and movie places. Many of them are there. How it works, is that you will see someone taking the drugs, using it then goes to chill at the pool, maybe go to the movies. So they can be found at the pool and movie places."* (FGD 2, respondent 5).

Five youths reported that males this age who use substances can be found in the community and in public places such as marketplaces and bus stages.

*"You can find them in marketplaces… You can find them hanging out and playing cards as they use the drugs in places they have created. They pour what they call jet fuel on fabric, sniffing it. You will find many of them with this." (FGD 2 respondent 6).*

Four youths reported that campus would be a good place to recruit males aged 18–24 years from. One said:

*"You can find them [on campus. You can approach them by holding] a conference and it shouldn't be boring. It should be something engaging…and then a prominent person can come. Music, food, something like that. My school loves free food so if you say there is an event with free food and gifts then they turn up." (FGD 1 respondent 1).*

Three youth reported that parties would be a good place to recruit males aged 18–24 years from. Two youths said that 18–24-year-olds who use substances could be found where substances are sold. One each reported that they could be found in the rescue center, barbershops and working places.

**3.4.5. Challenges to anticipate when recruiting youth for a substance use study. Schools:** Several youths reported that the fear of being reprimanded would form a barrier to recruitment in school. One said that:

*"The main reason [for not recruiting in schools] is because it is not a safe environment to talk and come forward. Some teachers know that this boy has a problem [with substances] and tries to do everything to send the boy home instead of talking to them." (FGD 2, respondent 1).*

**Hospitals:** One youth reported that the challenge with recruiting in the hospital is that youth rarely visit hospitals.

Ten youths reported that they had not visited a hospital over the past one year. Five youths reported that they had only visited the hospital once over the past one year.

**General challenges:** Several youths reported that hostility from youth would be a barrier to recruitment in a study evaluating a substance use intervention.

*"You might get insulted. Some of them might be rude. So, it is hard to approach them. Even if you approach them calmly, they may still insult you." (FGD 2, respondent 5).*

A few youths reported on other challenges, e.g., the youth may be disinterested in substance use counseling, may not give a true account of their substance use, and may not take the counseling seriously.

**3.4.6. Strategies for addressing identified challenges.** The youths proposed several strategies for overcoming these challenges including being friendly and creating a rapport with the youth, blending in with the youth, assuring them of confidentiality, and assuring them that there will be no repercussions if they disclose substance use. One youth said:

*"Also, you shouldn't dress very well when going to meet these people." (FGD 2, respondent 5). "One might judge you from a distance before giving you the information. So, you should be social and not too serious. Create friendship and they will open up." (FGD 2, respondent 3).*

*"Some people will withhold information… before this person can give you full information, they should really trust you. Someone will give you the full information depending on how you approach them. Theres also the dressing code." (FGD 2, respondent 2).*

**3.4.7. Feasibility of snowballing as a recruitment strategy.** Six youths reported that they knew more than one hundred youth who were using substances. Five youths reported that they knew about 10–50 people who were using substances, and only four knew less than 10 people who were using substances.

*"Mostly like my whole crew then people around me…over a hundred"* (FGD 3, respondent 2).

Most of the youth reported that they would be comfortable telling other youth about a substance use program.

*"[snowballing]…is an okay method because the youth talk to each other a lot. This [program] is a good thing. I might leave here and tell my friends about [the program], who will in turn tell others about [it]."*

*(FGD 1, respondent 4).*

A few youths had concerns about snowballing as a recruitment strategy. One youth reported that youth contacted through snowballing may be *"…shy and not confident to come… [may be] ashamed to come…"* (FGD 3, respondent 2). Another youth reported that they may be uncomfortable talking about the program with other youth. He remarked *"I won't be so comfortable talking about it. Not everyone will listen to what you have to say. They will begin saying that I don't please them."* (FGD 3, respondent 5).

**3.4.8. Ways to recontact youth for a follow-up session.** Most youths reported that it was best to recontact them via phone either through a call, Short Message Service (SMS), or WhatsApp text message. A few youths reported that for those without phones, a return date can be scheduled ahead of time so that they can attend the follow-up visit.

*"…some of us don't have phones. You just say on a certain date, at a certain time we can meet here. It can help us…"* (FGD 1, respondent 2).

### 3.5. Preliminary effect of SSBI on substance use, depression, anxiety, and quality of life

Table 11 shows the preliminary effects of SSBI on substance use, depression, anxiety, and quality of life, highlighting the SMD differences and 95% CIs between the SSBI and psychoeducation groups. The SMD for total ASSIST-Y scores was −0.33 (95% CI: −0.83 to 0.16), suggesting a small, non-statistically significant reduction in substance use favoring the SSBI compared with psychoeducation arm at month 3. Regarding ASSIST-Y scores for the most problematic

**Table 11. Preliminary effect (SMD/ Cohen's d) of intervention on substance use, mental health, and quality of life.**

| Variable | Study Arm | | Difference[2] | 95% CI[23] |
|---|---|---|---|---|
| | Control N = 32[1] | Intervention N = 38[1] | | |
| **Total ASSIST Y score (Endline – Baseline)** | −0.3 (10.5) | 3.3 (11.5) | −0.33 | −0.83, 0.16 |
| *Missing* | 2 | 5 | | |
| **ASSIST-Y score for most problematic substance (Endline – Baseline)** | −2.2 (5.2) | −0.7 (6.3) | −0.27 | −0.76, 0.23 |
| *Missing* | 2 | 5 | | |
| **Mean PHQ-9 Score (Endline – Baseline)** | 1.6 (5.2) | 0.5 (4.5) | 0.23 | −0.27, 0.72 |
| *Missing* | 2 | 5 | | |
| **Mean GAD-7 Score (Endline – Baseline)** | 1.7 (3.5) | −0.9 (3.9) | 0.70 | 0.19, 1.2 |
| *Missing* | 2 | 5 | | |
| **Mean WHO-QOL BREF Score (Endline – Baseline)** | −3.1 (11.6) | 2.1 (13.9) | −0.41 | −0.91, 0.09 |
| *Missing* | 2 | 5 | | |

1 Mean (SD).

2 Standardized Mean Difference between Control and Intervention.

3 CI = Confidence Interval.

substance, The SMD was −0.27 (95% CI: −0.76, 0.23), suggesting a small, non-statistically significant reduction in substance use favoring the SSBI compared with psychoeducation at month 3. For PHQ-9 scores, the SMD was 0.23 (95% CI: −0.27 to 0.72) suggesting a small, non-statistically significant reduction in depression scores in the psychoeducation group compared with the SSBI group at month 3. For GAD-7 scores, the SMD was 0.70 (95% CI: 0.19, 1.2), suggested a medium, statistically significant reduction in anxiety scores in the psychoeducation group compared with the SSBI group at month 3.

Finally, for WHO-QOL BREF scores, the SMD of −0.41 (95% CI: −0.91 to 0.09) indicated a moderate, but non-statistically significant, improvement in quality-of-life scores in the SSBI group relative to the psychoeducation group at month 3.

### 3.6. Fidelity

The mean fidelity scores were 33.82 out of 34 for the SSBI and 2.97 out of 34 for the psychoeducation (control).

## 4. Discussion

Our study provides a detailed account of the feasibility of conducting a future definitive trial to investigate the effectiveness of a peer-delivered SSBI for youth with moderate risk substance use. It was challenging to recruit youth aged 15–17 years with moderate risk substance use compared to the 18–24 age group, and we did not reach the goal sample size for this age group. This could be related to lower rates of substance use among the younger youth in Kenya [57]. Initially, we set out to recruit all youth, then randomize them to intervention or control afterwards. This approach resulted in loss to follow-up of 35% of the recruited youth and this negatively impacted on the study participation rate. During the second phase of recruitment, we changed the strategy and delivered the intervention or control at the point of recruitment. During this second phase, we made the decision to recruit only 18–24-year-olds.

In response to the challenges we met during recruitment, we made two adjustments to the protocol. First, we amended the protocol to allow us to conduct follow-up interviews using an online meeting platform for youth who had travelled out of town at the three-month follow-up. Secondly, we added a qualitative aim with the goal of exploring the best strategies for recruiting youth in a future trial. Youth reported that young people who use substances can be most effectively recruited from community settings, and that fear of being reprimanded could be a barrier to disclosing substance use in learning institutions. The challenges faced during this trial and the strategies employed to successfully overcome them provide valuable information that will be useful in designing a full-scale trial and trials exploring other youth interventions in SSA.

All the benchmarks for trial feasibility were achieved except the "proportion of participants meeting inclusion criteria who get excluded". This can be explained by the challenges with the study processes during phase 1 of recruitment specifically, the significant time lag between recruitment and randomization to either intervention or control.

Qualitative interview results identified several other places to recruit youth who use substances, challenges to anticipate when recruiting youth with substance use, and strategies for addressing them. These qualitative findings can inform future research involving youth who use substances in SSA.

The SSBI demonstrated a small positive preliminary effect on substance use. A future definitive trial is warranted to more clearly determine its effectiveness among youth.

**Key take aways from this pilot trial:**

1. It is best to randomize study participants to intervention or control at the point of consent to minimize loss to follow-up.

2. A combination of recruitment strategies is required to identify young people who use substances

3. A protocol for a future trial could include the possibility of conducting follow-up surveys via online methods. Alternatively, eligibility criteria should consider physical availability of participants for follow-up interviews.

## 5. Future work

Over the past few years our research work has focused on substance use BIs for youth in Kenya. We have explored feasibility and acceptability from the perspective of youth, peer-providers, and health care workers [32,37]. We have also explored the feasibility of a future trial (findings reported in this article). Several questions remain unanswered: Are BIs effective for substance use among young people in Kenya? What is the optimal dosing and content for BIs for youth in Kenya? Which youth sub-populations (e.g., age, gender) do the BIs work best for? What are the mechanisms of change for BIs among youth in Kenya? Which are the best settings to implement BIs for youth in Kenya? What are the perceptions of policy makers about BIs for youth in Kenya? Our future research will focus on addressing these key questions as we work towards solving the pressing problem of youth substance use in Kenya and SSA.

## 6. Limitations

One limitation of this study is that it was conducted in a single center. Challenges and solutions with recruitment strategies reported here may not be generalizable to the Kenya context. Secondly, we had baseline imbalances between the SSBI and psychoeducation groups likely due to the small sample size [58]. Accordingly, the SMDs from this study should be interpreted with caution, and a definitive trial with adequate sample size is needed to determine the effectiveness of the SSBI. Despite these limitations, our study provides information for other researchers interested in conducting similar research in Kenya and SSA.

## 7. Conclusion

This pilot study met most of the predefined minimum requirements for the feasibility criteria. Important lessons were learnt during this pilot that can be applied to a future full-scale trial. We can therefore conclude that it is feasible to conduct a RCT of a peer-delivered SSBI for youth with moderate risk substance use in Kenya.

## Supporting information

**S1 File. ASSIST-Y scoring.**
(DOCX)

**S2 File. Fidelity checklist.**
(DOCX)

**S3 File. CONSORT extension checklist.**
(DOC)

**S4 File. Codebook.**
(PDF)

**S5 File. Dataset A.**
(CSV)

**S6 File. Dataset B.**
(CSV)

**S7 File. Inclusivity in global research questionnaire.**
(DOCX)

**S8 File. Study protocol.**
(DOCX)

## Acknowledgments

We acknowledge the technical support received from the AMPATH mental health program while conducting this study.

## Author contributions

**Conceptualization:** Florence Jaguga, Matthew Turissini, Edith Kamaru Kwobah, Edith Apondi, Leslie A. Enane, Julius Barasa, Yvonne Olando, Mary A. Ott, Allan Kimaina, Matthew C. Aalsma.

**Data curation:** Florence Jaguga, Gilliane Kosgei, Matthew C. Aalsma.

**Formal analysis:** Matthew Turissini, Leslie A. Enane, Mary A. Ott, Matthew C. Aalsma.

**Funding acquisition:** Florence Jaguga, Matthew Turissini, Edith Kamaru Kwobah, Edith Apondi, Matthew C. Aalsma.

**Investigation:** Florence Jaguga, Matthew Turissini, Edith Kamaru Kwobah, Edith Apondi, Leslie A. Enane, Gilliane Kosgei, Matthew C. Aalsma.

**Methodology:** Florence Jaguga, Matthew Turissini, Edith Kamaru Kwobah, Edith Apondi, Leslie A. Enane, Yvonne Olando, Mary A. Ott, Allan Kimaina, Matthew C. Aalsma.

**Project administration:** Florence Jaguga, Matthew Turissini, Edith Kamaru Kwobah, Julius Barasa, Gilliane Kosgei, Allan Kimaina, Matthew C. Aalsma.

**Resources:** Florence Jaguga, Matthew Turissini, Matthew C. Aalsma.

**Supervision:** Florence Jaguga, Edith Kamaru Kwobah, Edith Apondi, Julius Barasa, Yvonne Olando, Mary A. Ott, Matthew C. Aalsma.

**Validation:** Florence Jaguga, Edith Kamaru Kwobah, Leslie A. Enane, Matthew C. Aalsma.

**Visualization:** Matthew C. Aalsma.

**Writing – original draft:** Florence Jaguga, Matthew Turissini, Edith Kamaru Kwobah, Edith Apondi, Leslie A. Enane, Yvonne Olando, Allan Kimaina, Matthew C. Aalsma.

**Writing – review & editing:** Florence Jaguga, Matthew Turissini, Edith Kamaru Kwobah, Edith Apondi, Leslie A. Enane, Julius Barasa, Gilliane Kosgei, Mary A. Ott, Allan Kimaina, Matthew C. Aalsma.

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
