## [Decision Letter · Decision Letter 0]

11 Dec 2025

Dear Dr. JAGUGA,

Thank you for submitting your manuscript to PLOS ONE. After careful consideration, we feel that it has merit but does not fully meet PLOS ONE’s publication criteria as it currently stands. Therefore, we invite you to submit a revised version of the manuscript that addresses the points raised during the review process.

We look forward to receiving your revised manuscript.

Kind regards,

Yasir Alvi

Academic Editor

PLOS One

Journal Requirements:

This project was funded, in part, with support from the Indiana Clinical and Translational Sciences Institute funded, in part by Grant Number UL1TR002529 from the National Institutes of Health, National Center for Advancing Translational Sciences, Clinical and Translational Sciences Award.

This project was funded, in part, with support from the Indiana Clinical and Translational Sciences Institute funded, in part by Grant Number UL1TR002529 from the National Institutes of Health, National Center for Advancing Translational Sciences, Clinical and Translational Sciences Award.

Reviewers' comments:

Reviewer's Responses to Questions

**Comments to the Author**

1. Is the manuscript technically sound, and do the data support the conclusions?

Reviewer #1: Yes

Reviewer #2: Yes

Reviewer #3: Yes

Reviewer #4: Partly

2. Has the statistical analysis been performed appropriately and rigorously?

Reviewer #1: Yes

Reviewer #2: Yes

Reviewer #3: Yes

Reviewer #4: I Don't Know

3. Have the authors made all data underlying the findings in their manuscript fully available?

Reviewer #1: Yes

Reviewer #2: Yes

Reviewer #3: Yes

Reviewer #4: Yes

4. Is the manuscript presented in an intelligible fashion and written in standard English?

Reviewer #1: Yes

Reviewer #2: Yes

Reviewer #3: Yes

Reviewer #4: Yes

Reviewer #1: A well conducted study. Comprehensive presentation has been made of both quantitative and qualitative data.

There is clear alignment with feasibility trial objectives.

Standardized and validated measures (PHQ-9, GAD-7, WHO-QOL BREF, ASSIST-Y) have been used.

There is methodological transparency in recruitment and fidelity monitoring.

This study makes a valuable contribution to adolescent substance use intervention research in low-resource settings and is methodologically rigorous and well-reported.

Reviewer #2: Method wise, the study is okay. Explanations are given appropriately. Just a minor change, In Table 4, write Number of participants outside brackets and percentages in brackets eg. 25 (78.1%) and NOT 78.1%(25). Make it uniform as shown in other tables.

Reviewer #3: Here is a list of specific comments. Note: line and page numbering in reviews and comments is based on ruler applied in Editorial Manager-generated PDF.

1. Page 2, lines 54–56: I recommend reporting their effect sizes and their 95% CIs.

2. Page 4, line 112: The term, brief interventions, is quite general. I recommend describing the brief interventions, especially the trial’s interventions, more specifically. [The brief intervention is not fully introduced until pages 13–14. I recommend briefly introducing them in the Introduction section.]

3. Page 8, lines 199–201: I recommend including the ASSIST-Y scores that are used to determine moderate and high risks. [After reading further, I understand that it is impossible to list the cutoffs for all 9 substances in the abstract. Please ignore the aforementioned comment and consider including a supplementary table listing the substance-specific cutoffs for moderate and high risks.]

4. Page 8, lines 203–204: Is this a subjective measure or an evident-based measure? If the latter, I recommend including relevant references.

5. Page 9, lines 206–207: I recommend describing which efficacy (secondary) outcomes will satisfy the standardized effect size of 0.2 in the target population.

6. Page 9, lines 227–228: I recommend specifying the block size and the age categories (I assume the stratification factor of age is not continuous).

7. Page 16, line 401: I recommend replacing “parametric” with ‘normally distributed.’

8. Page 18, Table 4: It is surprising to see many imbalance in baseline variables (Tables 4–7).

9. Figure 1:

(9a) The text in the first side table is truncated.

(9b) I recommend adding another box before Randomized for Eligible even if the number of eligible is the same as the number of randomized.

(9c) Given the detailed summary about the two-phase recruitment, I recommend separating the top two boxes by phases. Eligible participants in both phases are combined into the Randomized box.

Reviewer #4: line 87 and 104: referencing ahould be kept it uniform across paper

75 and 83 : is it BIs or BI..keep it uniform its used differently in introduction section

table 1: as paper is written in past tesnse, table should also follow that. Presently its in future tesnse

line 203:sample size calculation needs to be explanined more . from where n=50 came?

please mention inclusion and exclusion criteria under a single heading, at present its in different places.

Why was 2 phase recruitmrnt approach taken? from where n=72 came in?

table 1 ..why 3 month FU is blank for phase 1 and phase 2?

line 329...is using term clients suitable here?

line 649-652: how do we define these small/ moderate/ medium positive effects?? what is the definition?

overall: About analysis- can be presented in a better way. Looks little unclear at certain points.

**Do you want your identity to be public for this peer review?** For information about this choice, including consent withdrawal, please see our Privacy Policy

Reviewer #1: No

Reviewer #2: No

Reviewer #3: No

Reviewer #4: **Yes:** CHHOKAR RESHMI

---

## [Author Response · Author response to Decision Letter 1]

22 Jan 2026

Response: We have aligned the manuscript including author details and file naming with PLOS ONE's style requirements.

2. Please include a complete copy of PLOS’ questionnaire on inclusivity in global research in your revised manuscript.

Response: A completed version of the questionnaire has been provided as Supporting Information- Supporting File 6.

3. Thank you for stating the following financial disclosure: This project was funded, in part, with support from the Indiana Clinical and Translational Sciences Institute funded, in part by Grant Number UL1TR002529 from the National Institutes of Health, National Center for Advancing Translational Sciences, Clinical and Translational Sciences Award.

Response: The authors confirm that the funders had no role in study design, data collection and analysis, decision to publish, or preparation of the manuscript. The revised financial disclosure statement therefore reads as below. Please change it on the online submission on the authors’’ behalf:

“This project was funded, in part, with support from the Indiana Clinical and Translational Sciences Institute funded, in part by Grant Number UL1TR002529 from the National Institutes of Health, National Center for Advancing Translational Sciences, Clinical and Translational Sciences Award. The funders had no role in study design, data collection and analysis, decision to publish, or preparation of the manuscript." This amendment has been included in the resubmission cover letter.

4. Thank you for stating in your Funding Statement: This project was funded, in part, with support from the Indiana Clinical and Translational Sciences Institute funded, in part by Grant Number UL1TR002529 from the National Institutes of Health, National Center for Advancing Translational Sciences, Clinical and Translational Sciences Award. Please provide an amended statement that declares *all* the funding or sources of support (whether external or internal to your organization) received during this study, as detailed online in our guide for authors at http://journals.plos.org/plosone/s/submit-now. Please also include the statement “There was no additional external funding received for this study.” in your updated Funding Statement. Please include your amended Funding Statement within your cover letter. We will change the online submission form on your behalf.

Response: The authors have detailed all sources of funding and aligned the financial disclosure statement with journal requirements. Revised statement below. Please change it on the online submission on the authors’ behalf:

“This project was funded, in part, with support from the Indiana Clinical and Translational Sciences Institute (https://indianactsi.org/) funded, in part by Grant Number UL1TR002529 from the National Institutes of Health, National Center for Advancing Translational Sciences, Clinical and Translational Sciences Award. The Grant was awarded to M.A. The funders had no role in study design, data collection and analysis, decision to publish, or preparation of the manuscript. There was no additional external funding received for this study.”

Response: The ethics statement now only appears in the Methods section of the manuscript page 20

Response: This has been done see page 37-38

7. If the reviewer comments include a recommendation to cite specific previously published works, please review and evaluate these publications to determine whether they are relevant and should be cited.

Response: There is no requirement by the reviewers to cite specific previously published works.

Response: The authors confirm that the reference list is complete and correct

---

## [Decision Letter · Decision Letter 1]

24 Feb 2026

A pilot randomized controlled trial to explore the feasibility of a peer-delivered single-session brief intervention for youth with moderate risk substance use

PONE-D-25-52483R1

Dear Dr. JAGUGA,

We’re pleased to inform you that your manuscript has been judged scientifically suitable for publication and will be formally accepted for publication once it meets all outstanding technical requirements.

Kind regards,

Yasir Alvi

Academic Editor

PLOS One

Reviewers' comments:

Reviewer's Responses to Questions

**Comments to the Author**

Reviewer #3: All comments have been addressed

Reviewer #4: All comments have been addressed

2. Is the manuscript technically sound, and do the data support the conclusions?

Reviewer #3: Yes

Reviewer #4: Yes

3. Has the statistical analysis been performed appropriately and rigorously?

Reviewer #3: Yes

Reviewer #4: I Don't Know

4. Have the authors made all data underlying the findings in their manuscript fully available?

Reviewer #3: No

Reviewer #4: Yes

5. Is the manuscript presented in an intelligible fashion and written in standard English?

Reviewer #3: Yes

Reviewer #4: Yes

Reviewer #3: (No Response)

Reviewer #4: (No Response)

**Do you want your identity to be public for this peer review?** For information about this choice, including consent withdrawal, please see our Privacy Policy

Reviewer #3: No

Reviewer #4: **Yes:** Chhokar Reshmi

---

## [Editor Report · Acceptance letter]

PONE-D-25-52483R1

PLOS One

Dear Dr. JAGUGA,

I'm pleased to inform you that your manuscript has been deemed suitable for publication in PLOS One. Congratulations! Your manuscript is now being handed over to our production team.

Kind regards,

on behalf of

Dr. Yasir Alvi

Academic Editor

PLOS One